# Ketamine can reduce harmful drinking by pharmacologically rewriting drinking memories

Ravi K. Das [1]*, Grace Gale[1], Katie Walsh[1], Vanessa E. Hennessy[1], Georges Iskandar [2], Luke A. Mordecai[3], Brigitta Brandner[3], Merel Kindt[4], H. Valerie Curran[1] & Sunjeev K. Kamboj[1]

Maladaptive reward memories (MRMs) are involved in the development and maintenance of acquired overconsumption disorders, such as harmful alcohol and drug use. The process of memory reconsolidation - where stored memories become briefly labile upon retrieval - may offer a means to disrupt MRMs and prevent relapse. However, reliable means for pharmacologically weakening MRMs in humans remain elusive. Here we demonstrate that the N-methyl D-aspartate (NMDA) antagonist ketamine is able to disrupt MRMs in hazardous drinkers when administered immediately after their retrieval. MRM retrieval + ketamine (RET + KET) effectively reduced the reinforcing effects of alcohol and long-term drinking levels, compared to ketamine or retrieval alone. Blood concentrations of ketamine and its metabolites during the critical 'reconsolidation window' predicted beneficial changes only following MRM reactivation. Pharmacological reconsolidation interference may provide a means to rapidly rewrite maladaptive memory and should be further pursued in alcohol and drug use disorders.

[1] Clinical Psychopharmacology Unit, University College London, London, UK. [2] University College Hospital and University College Hospital at Westmoreland Street, London, UK. [3] Pain Management Centre, University College Hospital, London, UK. [4] Experimental Clinical Psychology, University of Amsterdam, Amsterdam, The Netherlands. *email: ravi.das@ucl.ac.uk

Overconsumption disorders such as harmful drinking, alcohol and substance use disorders (AUDs, SUDs), which represent leading causes of global preventable mortality and morbidity, are fundamentally acquired or learned behaviours[1]. Contemporary neuroscientific models posit that the adaptive reward learning processes that control motivated behavior can be usurped by addictive drugs[2] forging harmful drug-use behaviors that are encoded by maladaptive reward memories (MRMs)[3]. These MRMs are learned associations that encode the contingencies between drug-predictive environmental stimuli (e.g. the smell and taste of beer) and drug reward[4]. MRMs underlie the tendency of environmental trigger cues and contexts to grab attention and provoke motivated behavioral routines including craving[5], drug-seeking and excessive consumption. They are thus a core mechanism underlying alcohol overconsumption and long-term relapsing behavior that must be "unlearned" for curative amelioration of problematic drinking.

However, effective, targeted memory rewriting currently represents an unmet clinical challenge. Critically, once stabilized - or consolidated - into long-term memory storage, MRMs were thought to become long-lasting and essentially immutable, promoting rebound/ relapse even long after successful reduction or detoxification and abstinence[6]. Current treatments such as cognitive-behavioral or cue exposure therapy do not involve unlearning of MRMs[7], but rather, suppression by alternative learning. The continued latent existence of MRMs limits the long-term efficacy of these interventions and underlies the high relapse rates that typify AUD/SUDs[8,9].

Recent insights into long-term memory persistence and malleability may hold the key to directly rewriting maladaptive memories. Reconsolidation is a memory maintenance process whereby reactivated long-term memories temporarily destabilize in order to incorporate newly available information, and hence update their contents[10]. Preclinical research has shown that memory destabilization requires the right retrieval conditions. These are typically brief, cue-driven retrievals that incorporate novel information or prediction error[11] regarding outcomes. Once destabilized, memories rely upon an N-Methyl D-Aspartate Receptor (NMDAR) mediated—MAPK/ERK—protein synthesis cascade to reorganize the synaptic architecture encoding memory traces and restabilize or reconsolidate memories in their new form. By pharmacologically intervening with reconsolidation, it is theoretically possible to selectively target and weaken memories[12,13]. The temporary reconsolidation window of memory instability following reactivation therefore offers a unique and novel mechanism to directly rewrite MRMs and strip them of their relapsogenic potential at the source[14].

Reliable pharmacological MRM rewriting remains elusive, however, due to the relative difficulty in reactivating/destabilizing inherently robust MRMs in human drug users and the severely limited menu of well-tolerated reconsolidation blockers[15]. Indeed, most preclinical studies of reconsolidation involve experimentally generated "models" of MRMs that are orders of magnitude weaker than true human MRMs, and also employ highly toxic compounds (with highly limited human translatability) to block reconsolidation[16]. Thus, despite the great theoretical potential of reconsolidation as a therapeutic target and promising emergent research[17], in the absence of a gold standard reconsolidation blocker, the translational feasibility and scope of pharmacological memory rewriting remains relatively untested.

Ketamine is a dissociative anesthetic that may have unique potential in this regard, since it is a high-affinity non-competitive NMDAR antagonist that is relatively well tolerated and safe in humans. Ketamine is currently experiencing a renaissance in neuroscience and psychiatry due to its rapid and novel antidepressant action[18]. Further it has previously been used to

successfully treat alcoholism[19] and heroin addiction, via unexplored, but not explicitly reconsolidation-based mechanisms[20]. It thus carries potential therapeutic utility for addictive disorders in its own right. Importantly, these antidepressant and anti-addictive actions may not be independent, since depression and SUDs are highly co-morbid[21] and concomitant improvements in response to an anti-depressant intervention may be seen to the extent that the former is driving the latter. We therefore assessed for the first time whether intravenous ketamine during the 'reconsolidation window' would interfere with the reconsolidation of robust alcohol-MRMs in harmful drinkers by blocking NMDAR activity. To differentiate reconsolidation-dependent from non-specific affective (e.g. anti-depressive) therapeutic mechanisms, ketamine was administered following the retrieval/destabilization of maladaptive alcohol memories (retrieval + ketamine; RET + KET) or control (non-drinking) memories (No RET + KET), with placebo (saline; PBO) controlling for the effects of MRM retrieval per se (RET + PBO). We further assessed plasma ketamine and its metabolites during the critical "reconsolidation window" as potential predictive biomarkers of response to the memory-rewriting manipulation.

In RET + KET, we hypothesized that ketamine would weaken MRMs via reconsolidation interference, reducing the motivational effects of alcohol (alcohol/ cue reactivity) and drinking levels in hazardous/harmful drinkers. These changes should be negatively associated with levels of blood biomarkers of ketamine metabolism during the critical reconsolidation window, indicative of a reconsolidation-interference mechanism. We also predicted (smaller magnitude) improvement in these measures in No RET + KET, given the antidepressant and potential anti-AUD properties of ketamine alone, but that these would not be related to ketamine metabolite biomarkers following the memory retrieval and drug manipulation. No improvement was expected from MRM reactivation alone (RET + PBO). This three group design allowed us to differentiate competing mechanistic interpretations. If any effects of ketamine were purely due to anti-depressive effects and independent of memory reconsolidation, the retrieval manipulation should be inconsequential and no differential improvement trajectory should be observed between RET + KET and No RET + KET. We thus assessed reconsolidation as a novel potential therapeutic mechanism and a means for catalyzing the efficacy of ketamine in problematic drinking.

Here we report that MRM retrieval + ketamine produces a rapid reduction in the reinforcing and motivational properties of alcohol and substantial, lasting reductions in drinking levels compared to retrieval or ketamine alone. Plasma levels of ketamine and its metabolites are predictive of these beneficial effects only following MRM retrieval. These findings demonstrate MRM reconsolidation interference by ketamine and rewriting of reward structures surrounding alcohol. The subsequent, lasting clinical benefits observed suggest that this one-session intervention approach should be pursued in the future treatment of alcohol related disorders.

## Results

**Sample characteristics**. All in-text descriptive statistics represent mean ± SD. The sample were young-to-middle aged adults (age 27.5 ± 8.1 yrs). Despite lacking formal diagnoses of AUD nor seeking treatment, they had particularly high drinking levels (74.09 ± 37.92 UK units (8 g alcohol)/week) and AUDIT scores (22.13 ± 4.93), denoting physically harmful drinking and moderate-high risk of developing AUD. Participant characteristics for relevant variables are given in Table 1.

**Reactivity to alcohol**. Time (Day 1 vs. Day 10) × Group ANOVA found significant Time x Group interactions for "urge to drink"

**Table 1 Levels of drinking-related and demographic variables at baseline, with inferential tests**

| | RET + PBO | RET + KET | No RET + KET | Total | Statistic | p | FDR α |
|---|---|---|---|---|---|---|---|
| Age | 27.7 ± 8.33 | 26.5 ± 6.25 | 28.23 ± 9.57 | 27.48 ± 8.11 | 0.354 | 0.703 | 0.036 |
| Gender (N F/M) | 9 | 11 | 15 | 35 | $\chi^2_{(2)} = 0.27$ | 0.319 | 0.022 |
| N smokers | 18 | 19 | 19 | 56 | $\chi^2_{(2)} = 0.095$ | 0.954 | 0.047 |
| AUDIT C | 9.1 ± 1.03 | 9.2 ± 1.03 | 8.9 ± 1.18 | 9.07 ± 1.08 | 0.596 | 0.553 | 0.028 |
| AUDIT total | 22.17 ± 4.86 | 23.77 ± 5.15 | 20.47 ± 4.33 | 22.13 ± 4.93 | 3.559 | **0.033** | 0.003 |
| SCID | 0.93 ± 0.83 | 1.1 ± 0.92 | 0.77 ± 0.57 | 0.93 ± 0.79 | 1.344 | 0.266 | 0.019 |
| Motivation to reduce | 3.17 ± 0.59 | 3.13 ± 0.35 | 3.17 ± 0.34 | 3.16 ± 0.45 | 0.054 | 0.947 | 0.046 |
| OCDS obsession | 6.6 ± 3.31 | 6.57 ± 3.38 | 5.53 ± 3.45 | 6.23 ± 3.38 | 0.966 | 0.385 | 0.024 |
| OCDS compulsion | 11 ± 2.24 | 10.47 ± 2.66 | 10.93 ± 2.89 | 10.8 ± 2.59 | 0.371 | 0.691 | 0.033 |
| OCDS total | 17.6 ± 5.13 | 17.03 ± 5.52 | 16.47 ± 5.91 | 17.03 ± 5.49 | 0.315 | 0.731 | 0.038 |
| TLFB total units (last 14 days) | 135.2 ± 60.17 | 164.05 ± 86.82 | 130.04 ± 57.06 | 143.1 ± 70.16 | 2.098 | 0.129 | 0.008 |
| TLFB daily units (last 14 days) | 9.74 ± 4.28 | 11.76 ± 6.22 | 9.29 ± 4.08 | 10.26 ± 5.01 | 2.11 | 0.127 | 0.006 |
| TLFB total units (last 7 days) | 69.35 ± 32.21 | 86.73 ± 45.75 | 66.19 ± 32.08 | 74.09 ± 37.92 | 2.645 | 0.077 | 0.005 |
| N drinking days (last 14 days) | 11.07 ± 2.72 | 11.07 ± 2.78 | 11.2 ± 2.44 | 11.11 ± 2.62 | 0.025 | 0.975 | 0.050 |
| N drinking days (last 7 days) | 5.6 ± 1.33 | 5.63 ± 1.27 | 5.7 ± 1.47 | 5.64 ± 1.34 | 0.042 | 0.959 | 0.049 |
| N binge days (last 14 days) | 3.3 ± 3.31 | 4.73 ± 3.65 | 2.83 ± 2.32 | 3.62 ± 3.21 | 2.974 | 0.056 | 0.004 |
| N binge days (last 7 days) | 1.83 ± 1.79 | 2.6 ± 1.87 | 1.4 ± 1.35 | 1.94 ± 1.74 | 3.983 | **0.022** | 0.001 |
| SOCRATES Recognition | 24.87 ± 3.14 | 23.83 ± 3.42 | 23.33 ± 4.21 | 24.01 ± 3.64 | 1.4 | 0.252 | 0.017 |
| SOCRATES Ambivalence | 11.3 ± 2.9 | 10.97 ± 3.79 | 9.9 ± 3.45 | 10.72 ± 3.42 | 1.388 | 0.255 | 0.018 |
| SOCRATES taking steps | 21.77 ± 4.04 | 21.33 ± 4.04 | 20.87 ± 4.52 | 21.32 ± 4.18 | 0.343 | 0.71 | 0.037 |
| BAS drive | 11.1 ± 2.71 | 11.53 ± 2.98 | 10.57 ± 2.45 | 11.07 ± 2.72 | 0.951 | 0.39 | 0.027 |
| BAS fun | 13.27 ± 2.75 | 14.33 ± 1.6 | 13.77 ± 1.72 | 13.79 ± 2.11 | 1.957 | 0.147 | 0.009 |
| BAS reward | 16.63 ± 1.65 | 16.53 ± 2.62 | 16.3 ± 1.88 | 16.49 ± 2.07 | 0.201 | 0.819 | 0.041 |
| BIS | 20.87 ± 2.69 | 19.57 ± 2.81 | 19.97 ± 2.57 | 20.13 ± 2.72 | 1.837 | 0.165 | 0.013 |
| BDI | 15.47 ± 9.69 | 14.03 ± 8.58 | 11.3 ± 8.7 | 13.6 ± 9.07 | 1.657 | 0.197 | 0.014 |
| CEOA sociability | 25.77 ± 3.43 | 26.3 ± 4.16 | 24.43 ± 3.82 | 25.5 ± 3.86 | 1.904 | 0.155 | 0.012 |
| CEOA tension reduction | 7.3 ± 2.28 | 6.57 ± 1.81 | 7.03 ± 2.11 | 6.97 ± 2.07 | 0.96 | 0.387 | 0.026 |
| CEOA liquid courage | 12.93 ± 2.48 | 12.63 ± 2.98 | 12.23 ± 3.08 | 12.6 ± 2.84 | 0.453 | 0.637 | 0.031 |
| CEOA sexuality | 9.07 ± 2.8 | 8.3 ± 2.67 | 8.07 ± 2.43 | 8.48 ± 2.65 | 1.179 | 0.313 | 0.021 |
| CEOA impairment | 20.97 ± 5.24 | 21.3 ± 5.49 | 20.7 ± 3.98 | 20.99 ± 4.9 | 0.111 | 0.895 | 0.044 |
| CEOA Risk/aggression | 11.27 ± 3.04 | 12.03 ± 3.74 | 11.43 ± 3.46 | 11.58 ± 3.4 | 0.416 | 0.661 | 0.032 |
| CEOA self perception | 7.17 ± 2.37 | 6.77 ± 2.92 | 6 ± 2.23 | 6.64 ± 2.54 | 1.657 | 0.197 | 0.015 |
| DTS TOLERANCE | 2.92 ± 0.88 | 3.02 ± 1.01 | 2.93 ± 1.08 | 2.96 ± 0.98 | 0.091 | 0.913 | 0.045 |
| DTS ABSORBANCE | 2.79 ± 1.08 | 3.01 ± 1.12 | 2.92 ± 1.31 | 2.91 ± 1.16 | 0.273 | 0.762 | 0.040 |
| DTS APPRAISAL | 3.12 ± 0.96 | 3.55 ± 0.83 | 3.13 ± 1.09 | 3.27 ± 0.97 | 1.944 | 0.149 | 0.010 |
| DTS REGULATION | 3.1 ± 1.02 | 2.7 ± 1.08 | 2.88 ± 1.02 | 2.89 ± 1.04 | 1.115 | 0.332 | 0.023 |
| DTS TOTAL | 2.98 ± 0.81 | 3.07 ± 0.84 | 2.97 ± 0.98 | 3.01 ± 0.87 | 0.122 | 0.885 | 0.042 |
| PANAS +VE | 32.07 ± 7.91 | 33.47 ± 7.25 | 31.93 ± 8.11 | 32.49 ± 7.71 | 0.359 | 0.699 | 0.035 |
| PANAS −VE | 20.9 ± 6.38 | 21.1 ± 7.44 | 19.47 ± 6.91 | 20.49 ± 6.88 | 0.497 | 0.61 | 0.029 |

For continuous measures, the statistics given are mean ± SD, along with corresponding F values (DF 2, 87). For binary measures, Ns are given along with chi-square degrees of freedom and test values (denoted by $\chi^2$). "RET" = Retrieval, "KET" = Ketamine, "PBO" = Placebo. FDR α = False Discovery Rate alpha, calculated according to Benjamini and Hochberg[36]. P-values represent One-way ANOVA on group differences. Significant uncorrected p-values at p < 0.05 are highlighted in bold. Only p-values lower than the associated FDR α are significant at FDR-corrected p < 0.05. The groups did not differ on any baseline variables according to FDR criteria

ratings [$F(2, 87) \geq 6.489$, $p \leq 0.007$, $n_p^2 \geq 0.1$] (Fig. 1c, d and Supplementary Note 1), indicating significant reductions in $RET + KET$ in urge to drink a beer placed in front of them [$F(1,87) = 19.703$, $p < 0.001$, $n_p^2 = 0.185$] and post-consumption urge to drink more of the beer [$F(1,87) = 24.46$, $p < .001$, $n_p^2 = 0.219$] with no significant reduction in the control groups [$Fs < 0.5$, $ps > 0.48$]. Pre-drink anticipated enjoyment of beer also reduced in $RET + KET$ [$F(1,87) = 20.273$, $p < .001$, $n_p^2 = 0.189$] as did post-consumption actual enjoyment [$F(1,87) = 8.67$, $p = 0.004$, $n_p^2 = 0.091$]. Enjoyment did not change in the control groups [Time × Group interactions $F(2,87) = 8.234$, $p = 0.001$, $n_p^2 = 0.159$ and $F(2, 87) = 3.298$, $p = 0.042$, $n_p^2 = 0.07$, respectively] (Fig. 1a, b; Full detail and alcohol picture cue reactivity in Supplementary Note 1).

**Drinking behavior.** Subjective impressions of drinking changes showed significant *Group* effects for volume of drinking [$F(2,87) =$

3.164, $p = 0.047$, $\eta^2 = 0.07$], enjoyment of drinking [$F(2, 87) = 3.929$, $p = 0.028$, $\eta^2 = 0.08$] and general urge to drink [$F(2,87) = 5.071$, $p = 0.008$, $\eta^2 = 0.1$]. In all cases, this was driven by significantly greater reductions in $RET + KET$ than the other two groups [independent samples $ts > 2.36$, $p < 0.05$, $r > 0.29$ (individual tests in Supplementary Note 1)].

Linear mixed models on TLFB-rated number of drinking days/ week corroborated these ratings, with significant reductions in $RET + KET$ [$F(1,89.449) = 10.986$, $p = 0.001$, $n_p^2 = 0.084$], and no significant reduction in the control groups [Group × Time $F(2,89.85) = 3.802$, $p = 0.026$] (Fig. 2a). As participants may compensate for more days abstinent by drinking more/bingeing on drinking days, we assessed changes in total alcohol consumption and bingeing.

The $RET + KET$ group showed highly significant reductions in *general* alcohol consumption (beer, wine or spirits) from baseline to post manipulation [$F(1,89.17) = 19.55$, $p < 0.001$, $n_p^2 = 0.14$], equivalent to a reduction of 23.5 UK units/188 g ethanol over a

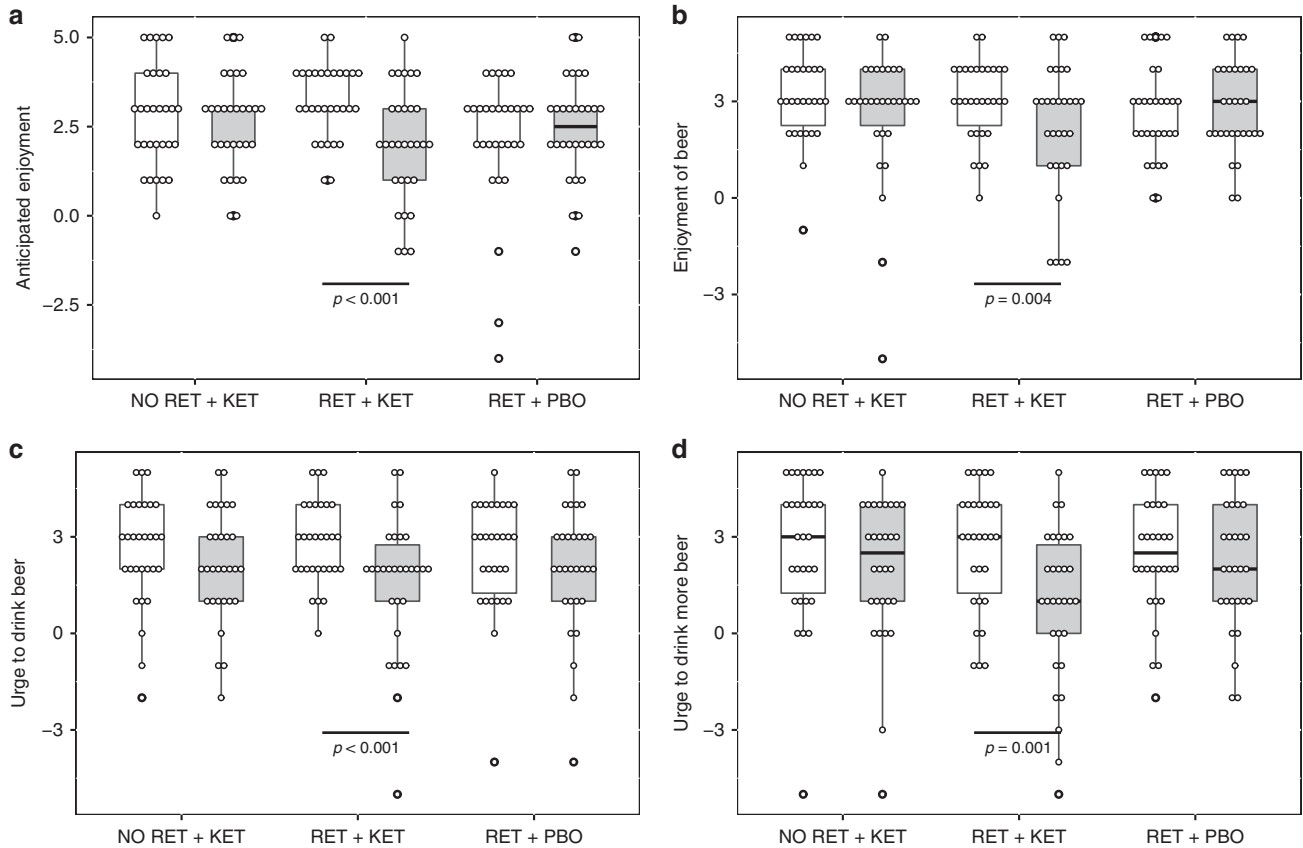

**Fig. 1** Reductions in motivational and reinforcing properties of beer in *RET + KET* from Day 1 (baseline) to Day 10 (post manipulation). **a** Anticipated enjoyment of beer. **b** Post consumption actual enjoyment, **c** Urge to drink beer, **d** Post consumption urge to drink more beer. In all cases, significant reductions were observed only in RET + KET. Boxes represent mean ± IQR, whiskers represent range. Dots are individual data points. *P* values represent *F* tests on multivariate simple-effects of Time within. Source data are provided as a Source Data file

week. A significant, although smaller, reduction was also seen in the *No RET + KET* group [$F(1,89.17) = 6.527$, $p = 0.012$, $n_p^2 = 0.052$], equivalent to a reduction of 13.6 UK units/109 g ethanol. No significant reduction in alcohol consumption was observed in *RET + PBO* [$F(1,89.95) = 0.726$ $p = 0.396$, $n_p^2 = 0.006$]; 4.9 UK units/39 g ethanol (Fig. 2b). The Group × Time interaction was marginally significant [$F$ (2,89.432 = 3.123, $p = 0.049$]. When achieved ketamine plasma concentration was taken into account (see Table 2, and "predictive biomarkers" section below) this interaction was further strengthened.

*RET + KET* also showed a highly significant reduction in binges (>6 drinks/week from baseline to post manipulation [$F(1,88.953) = 15.821$, $p < 0.001$, $n_p^2 = 0.116$], with no significant reductions in the control groups ($ps ≥ .22$, $n_p^2 ≤ 0.014$: trend-level Group × Time interaction $F(2,89.324) = 2.682$, $p = 0.074$]. Thus the *RET + KET* group were not compensating for reduced drinking frequency with greater drinking density.

**Long-term maintenance.** Reversion to heavy drinking typifies drinking interventions. We assessed this by comparing drinking levels post manipulation (Day 10) across follow-up to 9 months. Due to response attrition and missing data at each follow-up time point, linear mixed models were used to analyze follow-up data owing to better handling of missing data. Intercepts and slopes for Time (post manipulation, 2 week, 3, 6, 9 months) were modelled as random effects with an unstructured covariance matrix, due to improved fit over a fixed Time effect model ($-Δ2LL = χ^2(2) = 11.87$, $p = 0.002$). *Group* was included as a

fixed effect and baseline alcohol unit consumption as a covariate. This revealed further reductions in weekly alcohol consumption in all groups [Time main effect: $F(1,81.684) = 12.677$, $p = 0.001$], with no evidence of rebound to baseline levels (Fig. 2c), no further significant Group × Time effect was observed [$F(2, 81.54) = 0.091$, $p = 0.913$], indicating that the differential drinking reduction observed in *RET + KET* occurred rapidly following manipulation (by Day 10), with subsequent uniform reduction in all groups; consistent with a reconsolidation blockade effect. By 9 months, *RET + KET* had *halved* their average weekly consumption from ~84 to ~41 UK units. Figure 3 gives individual-level unit drinking data and distribution across all time points as pirate plots.

**Predictive blood biomarkers of response.** There is considerable inter-individual variation in the metabolism of ketamine, particularly in heavy drinkers where glutamatergic homeostasis is perturbed by chronic alcohol use. Table 2 shows Spearman rank correlations of post-infusion plasma ketamine levels and its metabolites norketamine (NK) and dehydroxynorketamine (dhNK) with primary outcomes. To the extent that reconsolidation blockade was the mechanism responsible for the observed reductions in drinking and that blood markers are a proxy for central ketamine availability, achieved plasma ketamine & metabolite levels during the "reconsolidation window" should predict subsequent drinking in *RET + KET*, but not *No RET + KET*. This is precisely what was observed, with moderate to large negative associations between ketamine levels and subsequent

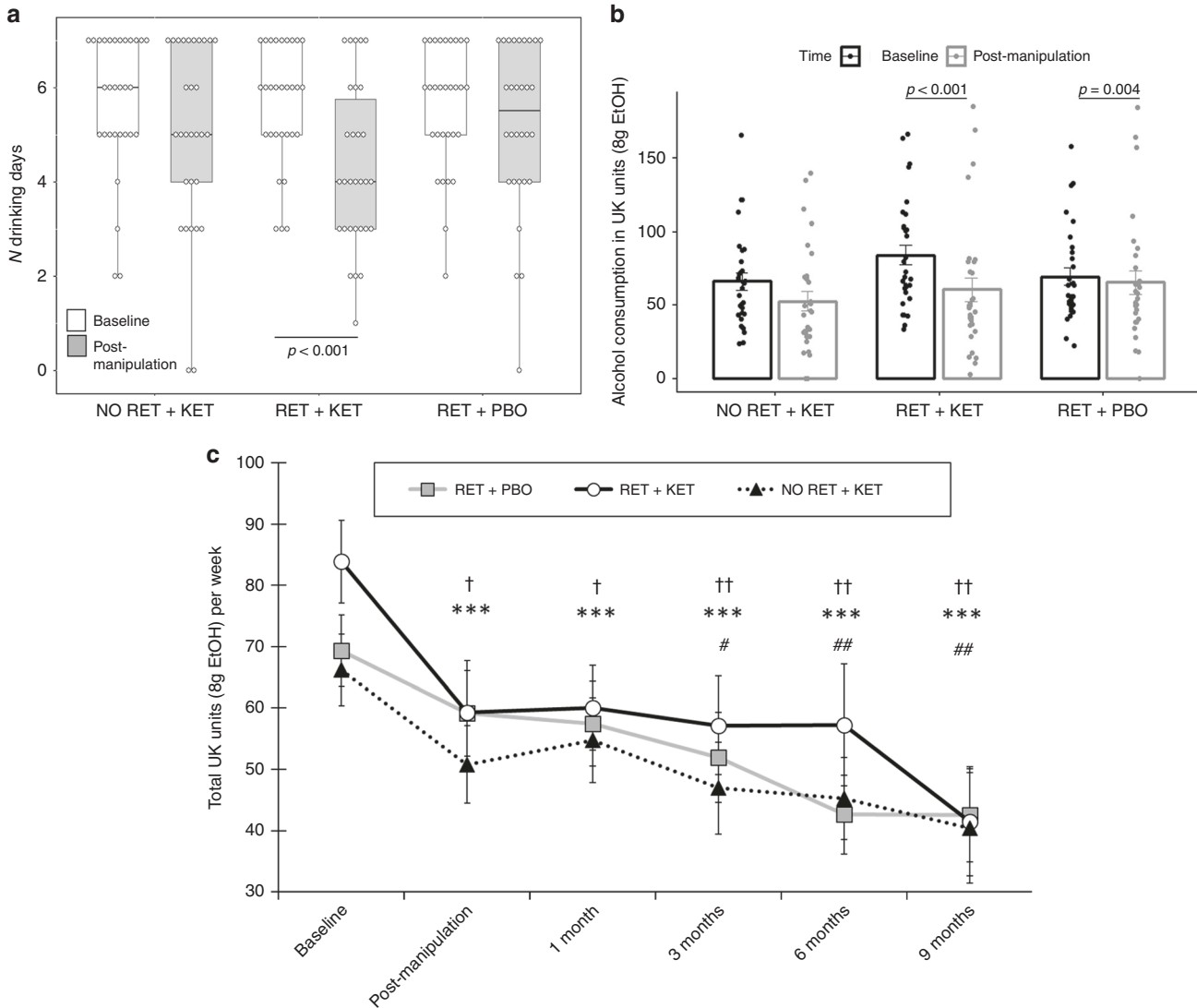

**Fig. 2** Changes in drinking outcomes. **a** Reduction in number of drinking days from Day 1 (baseline) to Day 10 (post manipulation) in RET + KET only. Dots are individual data points, boxes and lines represent mean ± IQR, whiskers represent range. **b** Reductions in total alcohol consumption per week (UK units) across the same period bars are mean ± SEM and dots individual data. **c** Changes in alcohol consumption across all measured time points. Data points and bars represent mean/SD using all participant data available at each time point. Significant reductions from baseline are denoted by † for No RET + KET, * for RET + KET and # for RET + PBO. One symbol = $p < .05$, two symbols = $p < .01$, three symbols = $p < .001$. In all panels, $p$-values are from F tests of simple contrasts against baseline within-group. Individual data points for **b**, **c** are given by group in Fig. 3. Source data are provided as a Source Data file

drinking in _RET + KET_ only. Intriguingly, the metabolites NK and dhNK better predicted craving and overall AUDIT scores, with dhNK uniquely predicting craving at 9 months.

Accounting for this variability in central ketamine concentrations by including achieved plasma ketamine levels as a covariate in the primary mixed-model analysis of unit alcohol consumption explained further variance in drinking, strengthening the Group × Day interaction [$F(2, 85.42) = 3.719, p = 0.037, \eta_p^2 = 0.078$].

## Discussion

This study found that intravenous ketamine following the brief retrieval of maladaptive cue-alcohol memories produced a comprehensive reduction in the reinforcing effects of alcohol among harmful drinkers. A rapid and lasting reduction in number of drinking days per week and volume of alcohol consumed was observed when ketamine followed MRM retrieval/destabilization, with no rebound to baseline observed for at least 9 months

following manipulation. Control groups receiving retrieval or ketamine alone did not show such changes in reward-related responses to alcohol, although the latter group did show some reduction in drinking.

This pattern of results is aligned with a therapeutic mechanism grounded in reconsolidation interference. Successful interference with the MRMs that putatively underlie excessive drinking should theoretically allow rapid and lasting dampening of reward responsivity to alcohol cues, reducing motivation to drink and drinking levels. The reductions in drinking attributable to ketamine per se (i.e. without MRM retrieval) are aligned with previous research indicating a potential therapeutic effect of ketamine in heavy drinking and addictive disorders, potentially via modification of glutamatergic dysregulation or mTOR-mediated downstream effects on neural plasticity[20]. Notably however, the effect of ketamine alone was considerably smaller than when combined with MRM retrieval. We therefore posit that prior MRM reactivation can be a potential catalyst for ketamine's

**Table 2 Spearman's rank correlations between ketamine metabolism and primary drinking outcomes post manipulation (Day 10) and at final follow-up (9 months) time-points**

| | | POST | | | | 9 MONTHS | | | |
|---|---|---|---|---|---|---|---|---|---|
| | | Drinking Days | Units consumed | Craving (ACQ) | AUDIT | Drinking Days | Units consumed | Craving (ACQ) | AUDIT |
| | | $N = 28$ | | | | $N = 19$ | | | |
| RET | Ketamine | **−0.465*** | **−0.543**** | −0.257 | −0.175 | −0.261 | **−0.449** | −0.149 | −0.44 |
| + | Norketamine | −.197 | −0.288 | −0.173 | **−0.422*** | −0.29 | **−0.481*** | 0.008 | **−0.627**** |
| KET | dhNK | −0.028 | −0.082 | **−0.518**** | **−0.457*** | **−0.458*** | −0.436 | **−0.572*** | **−0.573**** |
| | | $N = 29$ | | | | $N = 21$ | | | |
| No | Ketamine | −0.06 | −0.177 | −0.062 | −0.184 | −0.07 | −0.02 | 0.178 | 0.09 |
| + | Norketamine | −0.015 | −0.119 | −0.109 | −0.096 | −0.022 | −0.01 | 0.122 | −0.01 |
| KET | dhNK | −0.152 | −0.018 | 0.08 | −0.033 | −0.03 | −0.051 | 0.172 | −0.022 |

Significant correlations are highlighted in bold
*ACQ* alcohol craving questionnaire, *dhNK* dehydroxynorketamine
*$p < 0.05$, **$p < 0.01$, ***$p < 0.001$

efficacy in this scenario. Given the negligible additional time investment, discomfort, or clinical burden required to incorporate MRM reactivation, we recommend that this strategy is pursued to develop ketamine-based pharmacotherapies for AUD. This may further prove a fruitful approach in other disorders for which ketamine is currently under investigation and where maladaptive memory is implicated (e.g. depression and PTSD).

The moderate/large associations between blood ketamine and ketamine metabolite levels during the critical 'reconsolidation window' in $RET + KET$ are noteworthy, as they represent a potential biomarker for treatment response in a reconsolidation paradigm. That these associations were only seen in the "active" group strongly suggests that reconsolidation blockade was responsible for the remedial effects of the manipulation. Without prior destabilization of MRMs (No $RET + KET$), acute plasma levels of ketamine, norketamine and dehydroxynorketamine were relatively inconsequential to long term drinking levels. Since responding appeared dose-dependent and given that ketamine is relatively safe even at fully anesthetic doses, future studies may wish to consider using higher doses of ketamine (up to full anaesthesia) to maximize NMDAR saturation and subsequent memory interference.

These results are the first (to our knowledge) to demonstrate that reconsolidation of naturally acquired maladaptive alcohol memories in humans is dependent on NMDAR signaling, and that weakening of alcohol MRMs can be achieved with ketamine following MRM reactivation. The resultant, comprehensive reductions in cue reactivity and meaningful, lasting reductions in alcohol consumption outside of the lab after a single brief manipulation are unprecedented in alcohol research. This speaks to the potential scope of the reconsolidation-interference approach. Current "top-down" (psychosocial) treatment modalities that rely upon incremental learning of new, adaptive cognitive and behavioral patterns to suppress MRMs typically require prolonged treatment over multiple sessions. This presents issues both in terms of therapist burden and service user disengagement and recidivism.

The reconsolidation interference approach instead tackles this issue from the bottom-up, theoretically allowing direct weakening of pathogenic memory mechanisms and more rapid therapeutic gains. This is not to say the two approached need be mutually exclusive. Indeed the greatest treatment benefits may be seen through combination of an initial reconsolidation-based intervention to weaken relapsogenic memories, followed by cognitive-behavioral methods designed to instill more adaptive behaviors and cognitions.

Despite these promising results, several key issues remain that must be addressed through further study and refinement of this approach. Firstly, although ketamine is widely used and safe, particularly at the sub-anesthetic concentrations used here, its dissociative and psychotogenic properties and typical administration route (IV) mean specialist supervision is required and that it may be contraindicated for certain individuals with high schizotypal or dissociative traits. Contemporary advances in drug delivery technologies (e.g. intranasal) and the discovery of less dissociative analogs, spurred by ketamine's burgeoning use in depression, may be critical in improving the tolerability and acceptability of this approach in substance use disorders. Clearly, the tolerability and potential harms from single-dose ketamine (which we argue are minimal) must be weighed against the health benefits of reduced drinking. Drugs that act as antagonists/inhibitors of other pathways implicated in reconsolidation, such as noradrenergic antagonists may also hold promise for the weakening of maladaptive memories[22]. Although these remain relatively untested in the context of heavy drinking, meta-analysis suggests that these may be less generally effective in weakening reward memories than NMDAergic compounds[16].

Relatedly, although we suggest, based on preclinical research, that NMDAR antagonism is a likely potential mechanism underlying the observed effects, we cannot say with certainty that this is the only system involved in the current study. Ketamine has several targets, including other classes of glutamate receptor and opioid receptors which may have contributed to the observed effects. Although the NMDAR is thought to be the primary 'gatekeeper' of memory reconsolidation[23], non-NMDA receptors may also represent potential therapeutic targets for reconsolidation going forward.

A primary obstacle to the valid assessment of potential therapeutic reconsolidation-blockers is the lack of standardization in retrieval procedures designed to destabilize MRMs. Indeed, inconsistency in retrieval procedures is the norm in the field and may explain the inconsistency in studies attempting to interfere with memory reconsolidation[17,24–27]. We have attempted to address this issue through consistent use and detailed description of our MRM destabilization protocol[28]. However although effective, our procedure was not necessarily 'optimal'. Indeed, what constitutes 'optimal' retrieval parameters for destabilization of different memory types remains an empirical unknown that must be identified to realize the full potential of reconsolidation as a therapeutic strategy. Currently, when confronted with null results, we are unable to infer whether a failure to block memory reconsolidation, or a failure to destabilize memories a priori was responsible. This is due to the fact that memory destabilization and interference is currently a 'silent' process, lacking a valid biomarker. It must thus currently be inferred from successful reductions in behavioral "readouts" of MRM strength, as in the

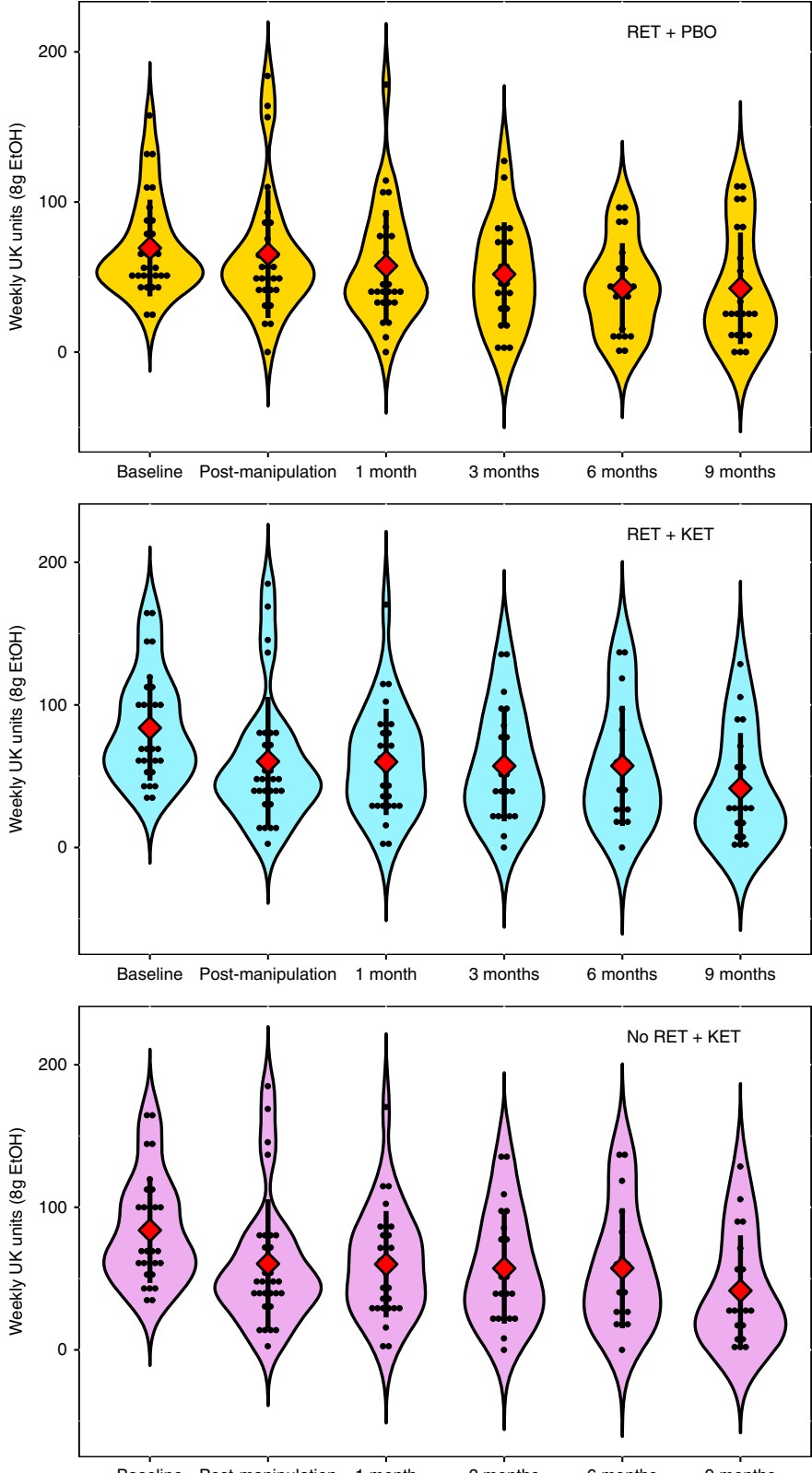

**Fig. 3** Individual-level total unit drinking data across all time-points. Top Panel = RET + PBO, Middle Panel = RET + KET, Bottom panel = No RET + KET. Dots represent individual data points and the "violins" the relative kernel density across the entire range of values. Diamonds and whiskers represent mean ± SD of each time point. Source data are provided as a Source Data file

current study. Thus despite the convergent evidence supporting this mechanism, we cannot say for certain that MRM weakening produced the beneficial effects observed here. Future research must tackle this issue directly, with the aim of developing independent biomarkers of memory destabilization. Having established ketamine as a robust, dose-dependent reconsolidation blocker in the current study marks a key step forward in achieving this aim and bringing this therapeutic approach to the clinic.

The participants in the current study showed a clearly harmful and problematic pattern of drinking, equivalent to that seen in clinical AUD, but had not received a formal diagnosis of AUD from a healthcare professional and were not treatment-seeking. There is significant variability in cut-point thresholds for diagnosing AUD from AUDIT scores in a UK drinking population. According to Foxcroft et al's[29] findings, based on mean AUDIT scores many of the sample might be expected to meet criteria for AUD. That the sample did not meet SCID criteria for severe alcohol dependence at screening is therefore noteworthy. This is because the sample scored very highly on measures of heaviness of consumption and effects of bingeing (which contributed greatly to AUDIT scores), but *did not* display physical symptomatology, extreme distress, inability to perform daily tasks nor morning drinking (which contribute highly to SCID criteria). These discrepancies raise important questions around exactly what is being assessed by alcohol use screening tools and potential response biases (see supplementary discussion). Given the novelty of the experimental manipulation assessed here, immediate assessment in a treatment-seeking sample would have been premature and carried greater potential for iatrogenic harm following a relatively untested intervention. Hazardous/harmful and non-treatment-seeking disordered drinkers are a key target group in their own right, however and the reductions observed here, could have enormous public health implications. Given the high levels of problematic drinking in the current sample, one may reasonably expect similar effects to be observed in a more severely dependent/ treatment-seeking population and there is now a strong rationale to conduct such clinical trials in formally diagnosed populations.

It is worth noting that baseline levels of alcohol consumption in *RET + KET* tended to be higher than the other two groups. While this difference was not statistically significant, we cannot rule out regression to the mean as a contributing factor to the observed reduction in alcohol consumption. Based on the pattern of results in their entirety, however, this explanation is highly unlikely. The clear and striking complementary reductions in the hedonic and motivational properties of alcohol, drinking frequency (which did not differ at baseline) and the association of these with objective ketamine biomarkers seen in *RET + KET*, are commensurate with the comprehensive dampening of alcohol reward memory structures that might be expected from successful MRM reconsolidation interference.

Owing to response attrition, power, and sample representativeness decreased throughout follow-up. Follow-up data showed that a self-selecting group of responsive participants. This may explain why the drinking data converge at the 9 month time point, with all groups reporting very similar (albeit much lower than baseline) levels of drinking. Despite this, intention-to-treat analyses did not show any appreciable difference to analyses performed on the available data.

This is the first study to demonstrate interference with the reconsolidation of maladaptive alcohol memories in humans using ketamine. These findings highlight the promise of reconsolidation interference as a therapeutic mechanism in harmful drinking, alcohol and substance use disorders and offers key insights into the therapeutic targets of ketamine, while adding to the burgeoning list of its potential psychiatric indications. The striking apparent dampening of reward structures surrounding alcohol and substantial, lasting reductions in drinking levels highlight that reconsolidation interference may form a key part utility of the next generation of more effective long-term treatments for addictive disorders.

## Methods

**Participants**. Participants were 90 beer-preferring men ($n = 55$) and women ($n = 35$) with hazardous/harmful drinking patterns, recruited via open internet advertisements. Despite a problematic pattern of drinking, participants did not have a formal diagnosis of AUD and were non-treatment seeking. Primary inclusion criteria were: scoring > 8 on the Alcohol Use Disorders Identification Test (AUDIT)[30]; not meeting SCID criteria for AUD at screening; Consuming > 40 (men) or > 30 (women) UK units/week (1 unit = 8 g ethanol), primarily drinking beer, non-treatment seeking (see Supplementary Methods).

**Design and procedure**. Ketamine infusion followed retrieval of alcohol-MRMs (RET + KET) or control (orange juice) reward memories (No RET + KET). A third group retrieved alcohol-MRMs prior to IV placebo (RET + PBO). Random allocation to the "active" group (RET + KET) and two control conditions (N = 30 per group) allowed us to assess effects of ketamine via reconsolidation, above those of ketamine per se. Drug manipulations were single-blind and placebo controlled. All participants completed a 3-day testing protocol at University College London (UCL) and the attached hospital (UCLH). Follow-up reassessment was performed up to 9 months. Attrition during remote follow-up left 9 month respondent Ns at: Ret + PBO = 20/RET + KET = 17/No RET + KET = 19. Participants were reimbursed for their participation. Written, informed consent was obtained prior to participation and all procedures were approved by the UCL Research Ethics Committee and UK Medicines and Healthcare Regulatory Authority, in line with the Declaration of Helsinki (2013).

We assessed clinically-relevant MRM weakening via (1) reactivity to sampled alcohol (beer) and alcohol cues (2) perceived changes in drinking levels, plus quantitative drinking days/week, binges/week and total alcohol consumption via the Timeline Follow-Back[31]. A three-day protocol was used. The first (Day 1) and final (Day 10) days provided "baseline" and "post manipulation" assessments of primary outcomes and questionnaire-based variables. Memory retrieval/ dug manipulation took place on Day 3. Procedure are registered under ISRCTN registry (No. 10138262, https://www.isrctn.com/ISRCTN10138262).

**Tasks and apparatus**. For cue reactivity assessment (Day 1 and Day 10), participants were given a 150 ml glass of beer and told they would consume this after rating a series of images. They then rated their induced urge to drink and liking of four orange juice images and four beer images (subsequently used as retrieval cues in RET/No RET procedures), plus three wine and two soft drink images (not used as retrieval cues), followed by their urge to drink the beer given to them and their predicted enjoyment of the beer. These were all on 11-point (−5 to +5) scales. They then consumed the beer according to timed prompts and rated their post-consumption actual enjoyment of the drink and urge to drink more. These scales thus assessed the hedonic and motivational properties of alcohol, which are central to excess consumption. These Day 1 procedures both allowed assessment of changes in cue reactivity and reinforcing properties of alcohol, and set the expectation of beer consumption to maximize PE when beer was withheld during reactivation on Day 3.

The MRM retrieval/destabilization procedure (Day 3) was one we have previously used to reactivate alcohol MRMs[32] and was identical to the cue reactivity task except (1) the beer was replaced with orange juice in the No RET + KET group (2) only four condition-appropriate cue images were rated (4 × orange juice images in No RET, 4 × beer images in RET groups) (3) in all groups, the drink was unexpectedly withheld at the appropriate timed prompt, generating negative prediction error, which has been shown to be a necessary condition for memory destabilization.[33] Ketamine hydrochloride or saline placebo infusion (I.V.) began 5 min after RET/No RET, procedures following a brief set of distractor tasks. Ketamine and placebo concentrations were maintained at 350 ng/dl for 30 min using a pharmacokinetic (domino) infusion model. Blood draws were taken 15 min pre and post infusion and gas chromatography was used to assay achieved plasma levels of ketamine, norketamine (NK) and dehydroxynorketamine (dhNK) and explore whether these, as a proxy for central concentrations during the "reconsolidation window", were predictive of responses to the manipulation.

On Day 10, participants repeated the cue reactivity task and reported perceived changes in their drinking behavior (volume, enjoyment and craving) since Day 1 using three five-point scales (+2 = greatly increased, −2 = greatly decreased). Drinking was quantified over the previous week on Day 1 ("baseline") and Day 10 ("post manipulation") via the Timeline Follow-Back[31]. Remote follow-up assessments of drinking (TLFB) were performed 2 weeks, 3, 6, and 9 months following Day 10 (see Supplementary Methods for full list of measures).

**Statistical approach**. Sample size was calculated in G*Power 3.1.9.2 for $1-\beta = 0.95$ to detect a minimum effect size of $n_p^2 = 0.05$ at $\alpha = 0.05$ for the interaction in 2 (baseline, post manipulation) × 3 (Group) mixed ANOVA, assuming $\rho$ of 0.5. This yielded a total required sample size of $N = 78$ (26 per group). Anticipating minimal attrition and technical error, we randomized $N = 30$/group.

Data analysis was performed using IBM SPSS 25 for Windows. Where sphericity was violated in repeated measures, the Greenhouse Geisser correction or multivariate terms were used, depending on ε values and according to the recommendations of Stevens[34]. Primary drinking-related dependent variables (cue reactivity, alcohol consumption), were assessed with 2 × 3 mixed ANOVA: within-subjects factor = Time (Baseline vs. post manipulation), between-subjects factor = Group (RET + PBO, RET + KET, No RET + KET). Significant $k > 2$ main effects and interactions in omnibus ANOVAs were investigated with multivariate simple effects analyses and paired tests on marginal means, where appropriate. Due to technical error, one participant's (male, RET + PBO) TLFB data were lost for the post manipulation time point. As such, these data and longer-term follow-up data on TLFB were analyzed using linear mixed models, including random intercepts per-participant, Group as a fixed factor and including participant-level random slopes across time if they improved model fit (assessed via Akaike's Information Criterion and chi-square tests on $\Delta -2LL$) and did not hinder convergence. For ANOVA, effect size is (partial) eta squared ($\eta^2/\eta_p^2$), was calculated by SPSS. For fixed effects in mixed models, pseudo—$\eta_p^2$ was calculated using the formulae from Westfall et al[35] Alpha for all a priori tests was set at 0.05, with $p$-values Bonferroni—corrected for post hoc tests. False discovery rate in analysis of baseline demographic variables was controlled with the Benjamini–Hochberg procedure[36]. All tests are two-sided. For full data handling, see Supplementary Methods.

## Data availability
The data that support the findings of this study are available from the corresponding author upon reasonable request. The source data underlying Figs. 1a–d, 2a–c, 3 and Supplementary Fig. 1 are provided as a Source Data file

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

## Acknowledgements
We would like to thank Professor Tony Absalom for his insights on ketamine infusion models and Dr. Kristian Warnes for his ongoing help formulating and dispensing the study drugs. This work was supported by the Medical Research Council (grant number: MR/M007006/1) to Drs. Das, Kamboj & Curran.

## Author contributions
R.K.D. designed the study, collected the data, analyzed the data, and wrote the paper. G.G., K.W., and V.E.H. collected and pre-processed the data. G.I., L.A.M., and B.B. provided medical oversight and performed drug infusions. M.K. and H.V.C. helped design and secure funding for the research. S.K.K. managed the research and edited the paper.

## Competing interests

The authors declare no competing interests.
