## [Peer Review File · Nature Communications]

Reviewers' comments:

Reviewer #1 (Remarks to the Author):

This is a very interesting ms. Presenting ketamine as a novel means to establish memory reconsolidation interference in problem drinkers with reduced drinking as a result. It is a well designed study and a well-written ms. I have a few points to consider for finalizing the ms.

1. Participants are described as "not diagnosed with AUD nor treatment seeking" (p. 17). The first part of this statement is most likely wrong, as the mean AUDIT score is above 20, which is at clinical levels. One recent UK source to consider is an Alcohol Insight publication, which validated the AUDIT as screener for DSM-IV and 5 diagnoses (Foxcroft et al 2015) and with a score of 22, the chance that participants meet DSM5 criteria for alcohol use disorder is above 80%. Hence, the first part of the sentence is technically correct if no structured diagnostic interview was done, but most likely incorrect for the large majority of participants. In the US this type of sample is often called "non-institutionalized AUD patients", which might be more correct. In any case, the current, somewhat misleading formulation should be adjusted.

2. Regarding the follow-up data, the authors use the often-used but invalid last-observation carried forward method (see Blankers et al 2010, J Medical Internet Res, for a demonstration in the same domain). In the same paper, it can be found that several multiple imputation methods do produce accurate results. I would therefore highly recommend that the clinical effects are assessed using this method.

Signed Reinout Wiers

Reviewer #2 (Remarks to the Author):

The investigators evaluated whether RET+KET was more effective than Non-RET+KET vs RET+PBO in unlearning MRM. Subjects with heavy alcohol use were randomized to one of these three interventions and subjects were followed for 9 months. Many measures were obtained including cue reactivity, liking, urge to drink, alcohol reactivity, and symptom ratings. The investigators found that RET+KET reduced the reinforcing effects of alcohol and long-term drinking patterns. Overall, this is an interesting experimental human study. Although the results are positive, there are several issues that need further clarification including methods, statistical approach (need to use linear mixed models), and consideration to other variables that might have influenced outcome.

Specific comments:

1. The introduction should include some of the competing or alternative hypotheses to MRM. For example, the antidepressant effects of ketamine is a possibility and the authors do show that. Negative affect was reduced only in the RET+KET group. It would be important to control for negative affect in the primary analyses. Also, ketamine produces anhedonia in healthy control subjects for up to 24 hours. It appears that the CADSS was higher in the RET+KET group than the Non-RET+KET group. Was this statistically significant. CADSS should be controlled for in the primary analysis.
2. Methods: urine drug screen at screen and follow-up? Do we know substances were not used on follow-up besides alcohol? Were concomitant medications permitted? If so, why? Prior use of ketamine?
3. There is no mention if other DSM/ICD diagnoses were permitted such as anxiety disorders, depressive disorders, etc.
4. Were lab tests performed to rule out medical conditions?
5. Blinding should be discussed. It is likely that raters and subjects were unblinded to the RET+PBO vs. KET groups. Were the raters who administered the RET or Non-RET the same as those who did the follow-up assessments? Was blinding maintained for RET or Non-RET procedures?
6. Clarify why an active control was not used such as midazolam. It is very likely that subjects easily determined whether they were in the RET+ PBO group. Was guessing the group assignment attempted by raters and subjects? If so, provide results, if not why was this not done?

7. How do we not that there was a "rewriting of the targeted memory"? I see there are measures of alcohol behaviors, but the outcome could have resulted from some non-specific effect, perhaps improvement in depressive or greater increases in anhedonic symptoms. In other words, I'm not confident that one can directly measure that there was a specific unlearning of MRMs. Is there a more direct biomarker to get at this to demonstrate, ie, central marker of target engagement?
8. Is it possible that the improvement was due to the greater effects of ketamine on depressive symptoms? The RET+KET had greater improvements in depressive symptoms than the Non-RET+KET group.
9. Next, the authors claim that NMDAR antagonist effects of ketamine was responsible for the findings (reconsolidation). However, there is no evidence provided that this was due to NMDAR antagonism. Ketamine has multiple off-target effects. Recent work suggests that its antidepressant effects are likely non-NMDAR antagonism. Could other targets be implicated i.e. AMPA, opioid? The levels of ketamine would not be sufficient as there are many ketamine metabolites that do not have NMDAR antagonist properties. The authors to not discuss this possibility.
10. Follow-up: the total UK units per week should be controlled for at base line. There is not really much difference at 9 months. It is likely if the baseline is controlled for that some of the results may no longer remain significant. Need to use linear mixed models here.
11. The analysis should include linear mixed models to account for the missing values rather than ANOVA and carrying the last observation forward.
12. Clarify if there was any difference in the CADSS in RET+KET vs. No RET+KET. If there was, this should be controlled for in the analysis.
13. How was the dose of ketamine selected? A dose finding study would have been much more informative.

Reviewer #3 (Remarks to the Author):

This is a timely study determining the capacity of i.v. ketamine to disrupt the reconsolidation of alcohol-associated memories that contribute to relapse risk in individuals showing hazardous drinking behaviour. While not in a clinical alcoholic population, this study is an important step to the clinical population, and demonstrates the value of the approach in a population at risk of developing alcoholism (i.e. as a preventative intervention).

The manuscript is well-written, and the experimental methods are described clearly (including sample size calculations and the methods used for statistical analyses). The analyses are all appropriate. I have only a few minor comments on the manuscript.

1. In the Introduction, the authors hypothesise that ketamine alone (No RET+KET) will produce small improvements in reducing the motivational effects of alcohol and drinking levels. It was not clear to me why this hypothesis had been drawn, and the reasoning behind it should be made more explicit.
2. A reference to Table 2 should be included in the paragraph of the results referring to the reduction in the number of binge drinking days in the RET+KET group.
3. In Figure 2C (and acknowledged by the authors in their Discussion) it is clear that the RET+KET group showed higher levels of drinking at baseline than the other experimental groups. Were the groups originally matched for drinking levels? If they were and the difference in baseline measures is due to participant dropout affecting this matching, it would be worth noting this in the manuscript.
4. Some of the references in the bibliography are missing details (e.g. volume and page numbers).

These minor points should be straightforward for the authors to address.

Amy L Milton

Das et al: Response to reviewers' comments

We would like to thank all reviewers for their positive, considered and insightful comments on the manuscript. We have attempted to make all requested changes to the manuscript, or otherwise further clarify our approach in our responses to each reviewer's comments in an itemized fashion below.

Reviewer #1

This is a very interesting ms. Presenting ketamine as a novel means to establish memory reconsolidation interference in problem drinkers with reduced drinking as a result. It is a well designed study and a well-written ms. I have a few points to consider for finalizing the ms.

Thank you to Prof. Wiers for these encouraging comments.

1. Participants are described as “not diagnosed with AUD nor treatment seeking” (p. 17). The first part of this statement is most likely wrong, as the mean AUDIT score is above 20, which is at clinical levels. One recent UK source to consider is an Alcohol Insight publication, which validated the AUDIT as screener for DSM-IV and 5 diagnoses (Foxcroft et al 2015) and with a score of 22, the chance that participants meet DSM5 criteria for alcohol use disorder is above 80%. Hence, the first part of the sentence is technically correct if no structured diagnostic interview was done, but most likely incorrect for the large majority of participants. In the US this type of sample is often called “non-institutionalized AUD patients”, which might be more correct. In any case, the current, somewhat misleading formulation should be adjusted.

Thank you for pointing this out. We agree that our description was somewhat unclear. The participants in the current study had not received a *formal diagnosis of AUD* from a healthcare professional and were not receiving treatment for such, but as Prof. Wiers points out, may well have met criteria for AUD given their high AUDIT scores. We have now clarified this description on p17 of the manuscript.

2. Regarding the follow-up data, the authors use the often-used but invalid last-observation carried forward method (see Blankers et al 2010, J Medical Internet Res, for a demonstration in the same domain). In the same paper, it can be found that several multiple imputation methods do produce accurate results. I would therefore highly recommend that the clinical effects are assessed using this method.

Having re-reviewed the cited literature, we are in agreement that LPCF can yield inaccurate estimates of treatment effects and may therefore be a sub-optimal approach to ITT analysis. We believe that in responding to *Reviewer 2's* comments below, we have addressed this issue. Briefly: We have now used a linear mixed-model approach to analyse these data. The mixed-model uses maximum likelihood estimation in modelling changes over time and thus affords better handling of this missing data that is less susceptible to bias than LPCF. These new analyses can be found in the results section on pages 12-13.

Reviewer #2 (Remarks to the Author):

1. The introduction should include some of the competing or alternative hypotheses to MRM. For

example, the antidepressant effects of ketamine is a possibility and the authors do show that. Negative affect was reduced only in the RET+KET group. I would be important to control for negative affect in the primary analyses. Also, ketamine produces anhedonia in healthy control subjects for up to 24 hours. It appears that the CADSS was higher in the RET+KET group than the Non-RET+KET group. Was this statistically significant. CADSS should be controlled for in the primary analysis.

The current study was designed to test a specific hypothesis about the therapeutic effects of ketamine *via blockade of MRM reconsolidation*, over and above non memory-specific (i.e. anti-depressant) effects. It was to differentiate these possibilities that the *No RET + KET* group was included, since this group isolates the *synergistic* effect of ketamine and memory retrieval/destabilisation, above ketamine alone. However, given the current level of interest in ketamine as an antidepressant, we agree with the reviewer that we should have been clearer about distinguishing these mechanisms from the potential for its anti-depressant actions to have knock-on effects on drinking. We have now expanded the introduction on pages 5 and 6 to highlight these alternative hypotheses and the patterns of response that would be expected favouring each alternative hypothesis.

We respond to the reviewer's suggestion regarding the CADSS data in response to their specific comments on dissociation (point 12) and suggestions regarding possible effects of ketamine on increasing anhedonia (point 7) and decreasing depression (point 7, point 8) in their relevant sections, below.

2. Methods: urine drug screen at screen and follow-up? Do we know substances were not used on follow-up besides alcohol? Were concomitant medications permitted? If so, why? Prior use of ketamine?

Use of other substances was based on self-report as we were unable to perform urinalysis to confirm drug use. We had stated in the supplementary materials that all participants were 'non treatment seeking' but appreciate that this was vague and have clarified that no CNS medications were permitted, nor was any recreational ketamine use. To this end, we have added further clarification on *Page 1* of the supplementary materials:

3. There is no mention if other DSM/ICD diagnoses were permitted such as anxiety disorders, depressive disorders, etc.

In the supplementary materials, the exclusion criteria previously included '*A diagnosis of AUD/SUD or other psychiatric disorder*'. To further reinforce this, we have amended this to say '*any other psychiatric disorder*' and added to the inclusion criteria '*non-treatment-seeking for AUD or any other psychiatric disorder*' [P1, supplementary materials].

4. Were lab tests performed to rule out medical conditions?

They were not, since exhaustive lab testing was beyond the financial scope of the current study. This is now acknowledged on *Page 1* of the *supplementary materials*.

5. Blinding should be discussed. It is likely that raters and subjects were unblinded to the RET+ PBO vs. KET groups. Were the raters who administered the RET or Non-RET the same as those who did the follow-up assessments? Was blinding maintained for RET or Non-RET procedures?

&

6. Clarify why an active control was not used such as midazolam. It is very likely that subjects easily determined whether they were in the RET+ PBO group. Was guessing the group assignment attempted by raters and subjects? If so, provide results, if not why was this not done?

Thank you for highlighting this omission. As the reviewer rightly notes, blinding was virtually impossible due to the pronounced subjective effects of ketamine. We have now added a section on blinding and condition guesses to *Page 3* of the *supplementary material*, along with statistical analysis of condition guesses and more information on blinding of follow-up data and analysis. It was impossible to blind the experimenters on the day of infusion due to the acute effects of drug and the necessity for different drinks to be given to the participants depending upon their MRM retrieval condition. However, follow-up data collection was performed by an experimenter blind to experimental condition and analysis was performed with a numerical code for group, the group identity being unknown to the analyst until primary data analysis was completed.

It is very difficult to actively control ketamine. There are clear differences between the subjective effects of midazolam and ketamine that are easily determined by experimenters with experience of giving both. Further GABAergic sedatives such as midazolam may themselves interfere with memory reconsolidation; as shown by Robinson and colleagues ¹ and Bustos and colleagues ². Using such a drug would have therefore not provided the experimental control we sought in the current study and further complicated the interpretation of the findings.

7. How do we not that there was a “rewriting of the targeted memory”? I see there are measures of alcohol behaviors, but the outcome could have resulted from some non-specific effect, perhaps improvement in depressive or greater increases in anhedonic symptoms. In other words, I’m not confident that one can directly measure that there was a specific unlearning of MRMs. Is there a more direct biomarker to get at this to demonstrate, ie, central marker of target engagement?

This is a central epistemic issue in many areas of psychology and cognitive neuroscience, not least in the reconsolidation literature. Memory interference, reconsolidation and updating are ‘silent’ processes, which are not directly observable. As there is currently no reliable biomarker for memory destabilisation and interference and we, like other reconsolidation researchers are limited to inferring reconsolidation mechanisms from behavioral manifestations and patterns of experimental effect. In that sense, the reviewer is entirely correct in saying we cannot ‘prove’ an unlearning of MRMs took place. This is not an issue that is specific to the current study, but for all studies of memory interference. A memory interference biomarker is something we are currently working on using neuroimaging metrics, but until such a biomarker is validated, this inference issue will remain.

We have now added to the discussion regarding the importance of developing tools for assaying the presence/absence of specific memories (Page 17 – 18).

Despite this, we feel that the pattern of results observed in the current study are most parsimoniously explained by a reconsolidation interference effect. Indeed, the outcome measures we selected (cue reactivity, drinking levels, reinforcing effects of alcohol) are thought to be rooted in MRMs and are thus secondary ‘indices’ or behavioural ‘readouts’ of MRM strength. We selected these measures to be able to triangulate such an effect and their convergence provides good evidence for reconsolidation interference. The three group design used here (where effects are observed in the retrieval + manipulation group) is the ‘gold standard’ for establishing reconsolidation interference and we observed patterns consistent with this interpretation. We are unsure what mechanism could more

parsimoniously explain this pattern of results. If, as the reviewer suggests, “...*the outcome could have resulted from some non-specific effect, perhaps improvement in depressive or greater increases in anhedonic symptoms*”, one would not expect differential outcomes in RET + KET and No RET+KET. We believe the reviewer may be referring to the work of Pomarol-Clotet³ and colleagues when invoking ketamine-related increases in anhedonia. In that study, ‘affective flattening’ was only observed in a subset of volunteers and was likely secondary to general sedative effects. Indeed in our group’s experience of giving ketamine to > 200 healthy volunteers, nor our work with recreational users, we have not observed significant anhedonic effects of acute ketamine. Further, we found no evidence for *increases* in anhedonia in the current study. Indeed, the SHAPS displayed quite the opposite. While acute effects were observed on *reducing* negative affect, these did not manifest in any change in depressive symptomatology as assessed by the BDI. Any change in affect was thus relatively low-level and we feel unlikely to be responsible for such clear changes in drinking and cue reactivity.

8. Is it possible that the improvement was due to the greater effects of ketamine on depressive symptoms? The RET+KET had greater improvements in depressive symptoms than the Non-RET+KET group.

We have partially addressed this comment in our response to points 1 and 7, in which the reviewer also forwards antidepressant activity as a mechanism for the observed results. To more fully flesh out our response here, we would like to highlight that *RET+KET* did not display greater improvements in *depressive symptoms* than *No RET + KET*, as no differential effects were observed on the BDI or SHAPS were observed. We believe the reviewer is referring to the differential effect on the negative affect scale of the PANAS. While the PANAS has been shown to correlate with measures of depressive symptomatology⁴, it was not designed as a diagnostic tool, nor to specifically index depression. The PANAS is a state measure and thereby sensitive to day-to-day variability in affect, rather than longer-term depressive symptomatology.

The reviewer raises an important point, however, that these negative affect changes could have contributed to changes in drinking levels. To assess this, we calculated PANAS change scores and correlated these with drinking levels at each time point. Neither in the sample as a whole, nor in *RET+KET*, or any other group, did these scores correlate with drinking at any time point. We have added these further analyses to the *supplementary materials* page 5-6:

9. Next, the authors claim that NMDAR antagonist effects of ketamine was responsible for the findings (reconsolidation). However, there is no evidence provided that this was due to NMDAR antagonism. Ketamine has multiple off-target effects. Recent work suggests that its antidepressant effects are likely non-NMDAR antagonism. Could other targets be implicated i.e. AMPA, opioid? The levels of ketamine would not be sufficient as there are many ketamine metabolites that do not have NMDAR antagonist properties. The authors to not discuss this possibility.

The reviewer rightly notes that ketamine has many non-NMDAR targets that could have contributed. Indeed, as we cannot assay NMDAR receptor binding and saturation *in vivo* in humans, we cannot say for certain that NMDAR antagonism was the pharmacological substrate of the observed effects. Our discussion of NMDAR mechanisms is based upon ample pre-clinical literature indicating the NMDAR as a key ‘gatekeeper’ of memory reconsolidation. While AMPAR and opioid targets may be implicated in the anti-depressant actions of ketamine, its antidepressant effects are *not* the actions under study here. Indeed preclinical research suggests AMPARs are less involved in

destabilization/reconsolidation than NMDARs⁵, with opioid receptor involvement remaining relatively under-studied.

Thus, although we cannot ascertain for certain the contribution of NMDA vs. other neurotransmitter systems to the observed effects, such fine-grained dissection of mechanism is unlikely to ever be achievable in an applied clinical setting and, we feel, therefore somewhat secondary to the demonstrated efficacy of ketamine as a reconsolidation-blocker with important behavioural effects.

To highlight these points, we have now added the further discussion of mechanistic interpretation to the discussion section (p17–18).

10. Follow-up: the total UK units per week should be controlled for at base line. There is not really much difference at 9 months. It is likely if the baseline is controlled for that some of the results may no longer remain significant. Need to use linear mixed models here.

There is a fundamental difference in the question being answered when modelling post-intervention (outcome) scores as a function of intervention, co-varying for baseline scores and in using a repeated measures/ mixed ANOVA approach. The former assesses between-group differences in outcomes when taking account of baseline differences, the latter assesses group differences in change in outcome, which was of primary interest in the current study. It is for this reason that we adopted a repeated measures, rather than baseline-adjustment approach.

However, in response to the reviewer's comment, we re-ran the analysis of primary outcome data (post-manipulation total unit consumption) as a function of *Group* while co-varying for baseline unit alcohol consumption. While an overall *Group* effect was not observed on *Day 10* [$F(2, 85) = 2.346, p = .102$] contrasts on marginal means revealed significantly reduced estimated alcohol consumption in *RET+KET* vs. *RET + PBO* on post-manipulation drinking [mean difference 16.81 units, $t(57) = 2.16, p = 0.034, r = .28$]. No significant difference between *No RET+KET* and *RET+PBO* was found [mean difference 9.28 units, $t(57) = 1.21, p = 0.23, r = .16$]. We have now added this analysis to the supplementary materials, adding to the 'Data Handling' section on page 4 and inserting the above requested analysis on supplementary page 5: We further address this comment in response to point 11, below where we re-run the follow-up analysis with LMMs, co-varying for baseline drinking.

11. The analysis should include linear mixed models to account for the missing values rather than ANOVA and carrying the last observation forward.

We thank the reviewer for this suggestion and agree that it is a superior approach. We have now re-analysed all the primary pre-post manipulation naturalistic drinking outcome data and secondary follow-up data using this approach; allowing random intercepts on a per-participant basis and modelling *Group* as a fixed effect, with random slopes for the *Time* effect where this did not lead to over-parameterization and non-convergent models. Use of this approach produced largely the same pattern of results as the RMANOVA analysis, although the results were marginally larger in favour of the described reconsolidation effect. The only substantial difference to note was that, in using the maximum-likelihood estimation/ LMM approach; the *Group X Time* effect on primary drinking outcome data (baseline to post-manipulation) became marginally significant at $p < 0.05$. While we are generally opposed to interpretation of results on the binary classifications of 'significance' and this represents a relatively small change in effect magnitude, we appreciate that this change may be 'significant' to some readers of the paper. The LMMs are now reported in place of the former univariate RMANOVA in the results section (pages 12- 13). We have also added to the 'Statistical Approach' section on page 10 and removed the LPCF analysis from the *supplementary materials*.

12. Clarify if there was any difference in the CADSS in RET+KET vs. No RET+KET. If there was, this should be controlled for in the analysis.

We thank the reviewer for this recommendation. The *RET+KET* group did in fact have a higher acute CADSS in response to ketamine while on-drug. We have now also included the analysis of CADSS scores specifically between the two ketamine groups in the supplementary discussion. As noted in the supplementary, there was no correlation between CADSS at any time point and any of the outcome measures. Given this, and that exploratory inclusion of CADSS on-drug score showed neither a covariate effect nor a Group X CADSS interaction, repeating the analyses with CADSS as a covariate was deemed inappropriate. We have this further analysis to the *supplementary material* (page 6).

13. How was the dose of ketamine selected? A dose finding study would have been much more informative.

Thank you for pointing out this omission, the dose was selected through piloting escalating doses starting with 200ng/dl used previously by our group and Pomarol-Clotet et al ³. We aimed for the highest dose for which we had ethical approval (350ng/dl) which would be well tolerated by participants. The highest allowable dose was well tolerated and selected for this reason. We have now added this information on dose selection to the supplementary materials (page 2).

Reviewer #3 (Remarks to the Author):

We thank Dr. Milton for her positive, kind and constructive comments on our manuscript. The minor amendments suggested to the manuscript have all been made and are itemised below.

1. In the Introduction, the authors hypothesise that ketamine alone (No RET+KET) will produce small improvements in reducing the motivational effects of alcohol and drinking levels. It was not clear to me why this hypothesis had been drawn, and the reasoning behind it should be made more explicit.

Thank you for pointing out this lack of clarity. The hypothesis was based upon prior human work from Evgeny Krupitsky demonstrating a therapeutic effect of ketamine in reduction alcohol consumption in the absence of an (explicit) reconsolidation-based mechanism. We have now added this information to the introduction section and expanded this discussion more generally (page 5).

2. A reference to Table 2 should be included in the paragraph of the results referring to the reduction in the number of binge drinking days in the RET+KET group.

This has now been added on page 12.

3. In Figure 2C (and acknowledged by the authors in their Discussion) it is clear that the RET+KET group showed higher levels of drinking at baseline than the other experimental groups. Were the groups originally matched for drinking levels? If they were and the difference in baseline measures is due to participant dropout affecting this matching, it would be worth noting this in the manuscript.

Allocation to groups was completely random and not matched by drinking levels in any way. Any difference in baseline drinking levels is thus entirely due to random chance and sampling error. This difference was therefore not due to dropout. We have discussed the randomization procedure in the *supplementary materials (page 1)*.

4. Some of the references in the bibliography are missing details (e.g. volume and page numbers).

Thank you for pointing these out, we have now been through the bibliography and updated the incomplete references (numbers 2,17,18,20,22,25,29 & 33 & supplementary 5, 6, 8, 9, 10,11,15, 19).

REFERENCES

1. Robinson MJF, Armson M, Franklin KBJ. The effect of propranolol and midazolam on the reconsolidation of a morphine place preference in chronically treated rats. *Front Behav Neurosci.* 2011;5.
2. Bustos SG, Maldonado H, Molina VA. Disruptive Effect of Midazolam on Fear Memory Reconsolidation: Decisive Influence of Reactivation Time Span and Memory Age. *Neuropsychopharmacology.* 2009;34(2):446-457. doi:10.1038/npp.2008.75
3. Pomarol-Clotet E, Honey GD, Murray GK, et al. Psychological effects of ketamine in healthy volunteers. *Br J Psychiatry.* 2006;189(2):173-179. doi:10.1192/bjp.bp.105.015263
4. Crawford JR, Henry JD. The Positive and Negative Affect Schedule (PANAS): Construct validity, measurement properties and normative data in a large non-clinical sample. *Br J Clin Psychol.* 2004. doi:10.1348/0144665031752934
5. Mamou C Ben, Gamache K, Nader K. NMDA receptors are critical for unleashing consolidated auditory fear memories. *Nat Neurosci.* 2006;9(10):1237-1239.

Reviewers' comments:

Reviewer #1 (Remarks to the Author):

I am generally satisfied with the answers and the ms.

Still one of my comments bugs my mind, also after seeing the revision: that is how to characterize the participants.

Under participants, it now says:

"Participants were 90 beer-preferring men (n=55) and women (n=35) with hazardous/harmful drinking patterns, recruited via open internet advertisements. Despite a problematic pattern of drinking, participants did not have a formal diagnosis of AUD and were non-treatment seeking."

They were included to score above 8 on the AUDIT (hazardous drinking), but their mean AUDIT score was 22, and in the discussion it is now stated that the chance that they meet a formal diagnosis is larger than 80%, based on the recent Audit study in the UK by Foxcroft and colleagues (now referred to in the discussion).

So we agree here, what I fail to understand is how apparently all of these participants "did not meet SCID criteria for severe alcohol dependence at screening". Isn't this totally bizarre? Based on a large recent UK study, at least 80% of participants should meet AUD criteria, given their AUDIT scores, and none of them meets SCID criteria. I think a remark concerning this remarkable discrepancy should be added to the discussion, and in the sentence under participants "according to the SCID" should be added. (which we now can conclude has little, if any value apparently).

Reviewer #2 (Remarks to the Author):

The reviewers have adequately addressed my concerns. I have no further concerns.

Reviewer #3 (Remarks to the Author):

The authors have addressed all of my previous comments, and I have no further comments to make on the manuscript.

Response to Reviewers: **NCOMMS-19-07661A**

On behalf of all authors, I would like to thank all the reviewers once again for sharing their time and expertise in consideration of the manuscript. Dr. Milton and Reviewer 3 are now satisfied with the manuscript and have requested no further changes. Our amendments are itemised below:

1. We have amended the figures to include participant-level dot plots of key outcomes as requested.
2. Thank you to Prof. Wiers for his insightful thoughts regarding characterising the sample. We are in complete agreement that they represent a particularly heavy-drinking sample and he is right to point out the discrepancy in SCID criteria and AUDIT. I feel that some insight into this issue is offered by the disparity in the nature of questions that are asked by the SCID and the AUDIT, respectively:

Participants scored particularly highly on AUDIT items pertaining to heaviness and frequency of drinking and bingeing (participants frequently scored 9-12 points on items 1 to 3 alone). Indeed, as confirmed by the TLFB data, their consumption was generally extremely high. They further scored highly on items assessing guilt/remorse, blackouts and injuries during drinking, very likely during heavy binge episodes. However, despite such drinking patterns, their general physical symptomatology (withdrawal, drinking despite problems), inability to complete daily required tasks (neglect of activities) and distress caused by drinking were not particularly high. Furthermore, they virtually never drank in the morning. The SCID is highly skewed to such measures of impairment and physical symptomatology. Clearly, the sample's concern with their drinking had never reached a sufficient level to seek treatment for AUD, since such behaviour would have exempted them from participation (we would like to highlight that potential participants *were* excluded at screening based on their SCID score, further contributing to the current sample's characteristics).

There is no shortage of drinkers such as these in the UK; i.e. whose *consumption* levels would meet those of clinical criteria, but within their sociocultural milieu, do not see themselves as 'alcoholic', nor find their drinking overly impacts upon their daily function.

It is further feasible that the participants answered the AUDIT questions more truthfully, as they pertain to what might be perceived as less 'medical/clinical' aspects of alcohol use, but downplayed their responses to the SCID owing to the clear association with physically evident 'alcoholism' and the ensuing ego threat.

I feel that this raises important questions about the impact of culture-specific normative expectations upon the validity of different screening/diagnostic tools, the utility of universal cut-offs, what *exactly* is being assessed by the AUDIT and the SCID and *how* they are answered. This is a thorny issue, to which a whole paper could be dedicated. Indeed, we would be very interested to discuss writing such a paper with Prof. Wiers. Full discussion is therefore beyond the scope (and allowed word count) of the current manuscript, although we have highlighted the key issues in the discussion section of the manuscript and further in the supplementary material. These are highlighted in green on the revised manuscript.

REVIEWERS' COMMENTS:

Reviewer #1 (Remarks to the Author):

I am satisfied with the authors' response and comments in the discussion and agree the discrepancy between audit and scid is worth a further commentary and or diagnostic research project.

My congratulations with very nice publication.

Atb
Reinout Wiers

Reviewer #2 (Remarks to the Author):

The authors have addressed my concerns. I have no further concerns.

Reviewer #3 (Remarks to the Author):

The authors addressed my points on the previous revision of the manuscript, and I have no further comments.